# Gut-directed hypnotherapy versus standard medical treatment for nausea in children with functional nausea or functional dyspepsia: protocol of a multicentre randomised trial

Pamela D Browne,[1] Bibiche den Hollander,[1] Esther M Speksnijder,[1] Herbert M van Wering,[2] Walther Tjon a Ten,[3] Elvira K George,[4] Michael Groeneweg,[5] Nanja Bevers,[6] Margaretha M S Wessels,[7] Maartje M van den Berg,[8] Joery Goede,[9] Sarah T A Teklenburg-Roord,[10] Carla Frankenhuis,[1] Marc A Benninga,[1] Arine M Vlieger[11]

For numbered affiliations see end of article.

**Correspondence to**
Dr Pamela D Browne;
p.d.browne@amc.uva.nl

## ABSTRACT

**Introduction** The treatment of chronic functional nausea or nausea due to functional dyspepsia in children is generally symptomatic. Moreover, these disorders pose a risk for worse psychosocial and health outcomes in children. Hypnotherapy (HT), by its ability to positively influence gastrointestinal and psychosocial functioning, may be an effective treatment for chronic nausea.

**Methods and analysis** To test efficacy, this multicentre, parallel, randomised controlled, open label trial evaluates whether gut-directed HT is superior to standard medical treatment (SMT) for reducing nausea. The study will be conducted at eleven academic and non-academic hospitals across the Netherlands. A total of 100 children (8–18 years), fulfilling the Rome IV criteria for chronic idiopathic nausea or functional dyspepsia with prominent nausea, will be randomly allocated (1:1) to receive HT or SMT. Children allocated to the HT group will receive six sessions of HT during 3 months, while children allocated to the SMT group will receive six sessions of SMT+supportive therapy during the same period. The primary outcome will be the difference in the proportion of children with at least 50% reduction of nausea, compared with baseline at 12 months' follow-up. Secondary outcomes include the changes in abdominal pain, dyspeptic symptoms, quality of life, anxiety, depression, school absences, parental absence of work, healthcare costs and adequate relief of symptoms, measured directly after treatment, 6 and 12 months' follow-up. If HT proves effective for reducing nausea, it may become a new treatment strategy to treat children with chronic functional nausea or functional dyspepsia with prominent nausea.

**Ethics and dissemination** Results of the study will be publicly disclosed to the public, without any restrictions, in peer-reviewed journal and international conferences. The study is approved by the Medical Research Ethics Committees United (MEC-U) in the Netherlands.

**Trial registration number** NTR5814.

## Strengths and limitations of this study

► First study to investigate the effectiveness of hypnotherapy on symptoms of nausea in children and adolescents, diagnosed with chronic idiopathic nausea or functional dyspepsia.

► A multicentre study with eleven participating academic and non-academic hospitals recruiting 100 children and adolescents.- Long-term study follow-up of 1 year.

► Due to the nature of hypnotherapy, children, parents and healthcare providers are not blinded for the received treatments.

## INTRODUCTION

Chronic idiopathic nausea (CIN) and functional dyspepsia (FD) affect approximately 0.5% and 4.5%–7.6% children worldwide,[1] respectively, and are associated with substantial physical and psychosocial distress, school absences and decreased social functioning.[2–4] Moreover, it has a considerable negative financial impact on healthcare.[5] According to the Rome IV criteria, when no evidence of organic disease is found, the disorders are considered functional. Children meet the Rome IV criteria for CIN when they suffer from chronic nausea without abdominal pain, when symptoms are not related to meals, and not consistently associated with vomiting. Children are diagnosed with FD when they have chronic symptoms of epigastric pain/burning, symptoms of postprandial fullness and/or early satiation.[6]

The treatment of CIN and FD with prominent nausea in paediatric patients is mostly symptomatic and not well defined. Most

clinicians individualise the patient's medical treatment, including prokinetics, antiemetics, antacids and herbal products, according to their symptoms and associated comorbidities.[3 4] The major disadvantage of this approach is that this treatment is symptomatic, and thus drugs often need to be used as long as patients suffer from nausea, which may take years.[7 8] Hence, there is a need for additional effective treatments for nausea in children with CIN or FD.

Several pathophysiological mechanisms have been proposed to play a role in the aetiology of CIN and FD, including delayed gastric emptying, impaired gastric motility and/or abnormal central nervous system processing of gastric stimuli through the gut–brain axis.[3] Additionally, there are indications that psychological factors, including anxiety and stress, may increase the severity of nausea (SON) through the gut–brain axis.[9 10]

Gut-directed hypnotherapy (HT) may have the potential to reduce symptoms of nausea in children with CIN or FD. HT is a form of therapy in which a therapist, by using suggestions, can induce a hypnotic state in an individual to positively modify physiological, cognitive and affective processes, as well as behaviour in that individual.[11] It has been shown to be very effective in the treatment of adults and children with functional abdominal pain[12 13] and children with chemotherapy-induced nausea and vomiting.[14] Therefore we hypothesise that HT, by its ability to influence gut motility,[15] psychological well-being[16] and visceral hypersensitivity,[17–19] might alleviate symptoms of nausea in children with CIN or FD as well. To date, however, no studies have examined the potential effect of HT in children with CIN or FD.

The main goal of this multicentre randomised controlled trial (RCT) is to evaluate the effectiveness of HT in reducing symptoms of nausea in children with CIN or FD. Six sessions of gut-direct HT will be compared with six sessions of standard medical treatment (SMT) plus supportive therapy in 100 children with CIN or FD between 8 and 18 years. Additionally, we will investigate the potential influence on abdominal pain, dyspeptic symptoms, quality of life (QoL), anxiety, depression, school absences, parental absence of work and healthcare costs. We hypothesize that HT will be more effective in reducing symptoms of nausea than SMT. Furthermore, we expect that children receiving HT will report more relief of symptoms (eg, less abdominal pain, less dyspectic symptoms), better QoL, less symptoms of anxiety and depression, less absence from school, compared with children receiving SMT. We also expect that parents of children in the HT group will report less parental absences from work and lower healthcare costs, compared the medical treatment group.

## METHODS
### Trial design
The present study in children and adolescents is a multicentre RCT. One hundred children between ages 8 and 18 years with symptoms of nausea and fulfilling the Rome IV criteria for CIN or FD, diagnosed by their paediatrician, will be enrolled in the study. After randomisation, children will receive either six sessions of gut-directed HT during 3 months by a qualified hypnotherapist, or six sessions of SMT plus supportive therapy from their paediatrician during 3 months (see figure 1). Detailed information on the HT and SMT interventions can be found under the Intervention section. The online supplementary additional file 1 presents the Standard Protocol Items: Recommendations for Interventional Trials checklist (see online supplementary additional file 1).

### Patient and public involvement
Patients and the public were not involved in the design of the RCT.

### Recruitment
#### Recruitment procedures
Children and adolescents will be recruited in outpatient paediatric clinics of 1 academic and 10 non-academic hospitals in the Netherlands: Amsterdam University Medical Center (Amsterdam), Amphia Hospital (Breda), Maxima Medical Center (Veldhoven), Northwest Clinics (Alkmaar), Maasstad Hospital (Rotterdam), Zuyderland Medical Center (Heerlen), Rijnstate Hospital, (Arnhem), Haaglanden Medical Center (Den Haag), Spaarne Hospital (Hoofddorp), Isala Clinics (Zwolle) and St. Antonius Hospital (Nieuwegein). These centres are located in both urban and rural areas throughout the Netherlands, serving an ethnically diverse paediatric population.

#### Participant screening
All children with symptoms of nausea and fulfilling the Rome IV criteria for CIN or FD, will undergo blood laboratory testing before inclusion, including complete blood cell count, C-reactive protein, liver function tests, creatinine, total bilirubin and for celiac screening, amylase anti-transglutaminase antibodies and IgA testing. Additionally, urinalysis and stool analysis for parasites (*Giardia Lamblia, Entamoeba Histolytica*) and *Helicobacter pylori* antigens will be performed. The need for additional diagnostic testing, for example endoscopy to rule out eosinophilic oesophagitis or 24-hour pH, will be left to the discretion of the treating paediatrician or paediatric gastroenterologist. The flow of the study protocol is presented in figure 2.

### Criteria
#### Inclusion criteria
A total sample of 100 children and adolescents with CIN or FD with symptoms of nausea will be enrolled in the study. Children and adolescents can participate in this study if they meet the following inclusion criteria:
► Age 8–18 years at inclusion of the study.
► Diagnosis of CIN or FD, with symptoms of nausea, according to Rome IV criteria.[6]
► Sufficient knowledge of the Dutch language.

| TIMEPOINT | Enrolment | Allocation | During treatment | | Follow-up | |
| --- | --- | --- | --- | --- | --- | --- |
| | Before treatment (T-1) | Start of treatment (T0) | 6 weeks after the start of treatment (T1) | 3 months after the start of treatment (T2) | 6 months follow-up (T3) | 12 months follow-up (T4) |
| **ENROLMENT:** | | | | | | |
| **Eligibility screen** | X | | | | | |
| **Informed consent** | X | | | | | |
| **Randomisation** | X | | | | | |
| **INTERVENTIONS:** | | | | | | |
| **Hypnotherapy group (HT)** | | ●———————————● | | | | |
| **Standard medical treatment group (SMT)** | | ●———————————● | | | | |
| **ASSESSMENTS:** | | | | | | |
| Nausea and abdominal pain | X | | X | X | X | X |
| Dyspeptic symptoms | | X | | X | X | X |
| Health related quality of life | | X | | X | X | X |
| Anxiety and Depression | | X | | X | X | X |
| Cost effectiveness/cost utility | | X | | X | X | X |
| Work absenteeism by parents and school absenteeism by children | | X | | X | X | X |
| Somatisation | | X | | X | X | X |
| Adequate relief | | | | | X | X |

**Figure 1** Standard Protocol Items: Recommendations for Interventional Trials figure displaying the trial design and the outcome measurements. After screening for eligibility (T-1), children and parents sign the informed consent form and fill in the baseline questionnaire (T0). Children are then randomised in the hypnotherapy or standard medical treatment group. Assessments take place before the start of treatment (baseline; T0), 6 weeks after the start of treatment (T1), 3 months after the start of treatment (T2), 6 months' follow-up (T3) and 12 months' follow-up (T4).

## Exclusion criteria

Children will not be enrolled in the study if they meet the following exclusion criteria:

► Concomitant organic gastrointestinal disease.
► Simultaneous treatment by another healthcare professional for symptoms of nausea.
► Previously received HT.
► Intellectual disability.

## Randomisation, blinding and treatment allocation

After obtaining informed consent, children are randomly allocated, by the treating paediatricians, to one of the two treatment arms: HT, given by a qualified therapist, or SMT, meaning treatment by the child's paediatrician. A computerised random-number generator will be used to randomly allocate children on a 1:1 basis with varying block sized of 2, 4 and 6. To ensure allocation concealment, central randomisation will be applied and the random allocation sequence remains concealed from paediatricians enrolling children into the study. Due to the nature of HT, it is not possible to blind the participating children and healthcare professionals involved in the treatment of the participants.

## Intervention

### Hypnotherapy

Individual HT consists of six sessions of 50–60 min, given over a period of 3 months by a qualified hypnotherapist (weeks 1, 2, 3, 5, 7 and 11). Twelve hypnotherapist affiliated to the recruiting hospitals will offer the HT to children. All hypnotherapists have years of experience in performing HT in children. The hypnotherapists will use an adapted version of our previously used HT protocol.[13 20] The HT protocol contains exercises focusing on normalisation of the gut motility, stress reduction and ego strengthening. The hypnotherapists will be instructed to use the same scripts, but are allowed to adapt the content to the child's needs. The same protocol is used for children of all ages. However, the language used will be adjusted to the child's developmental age.

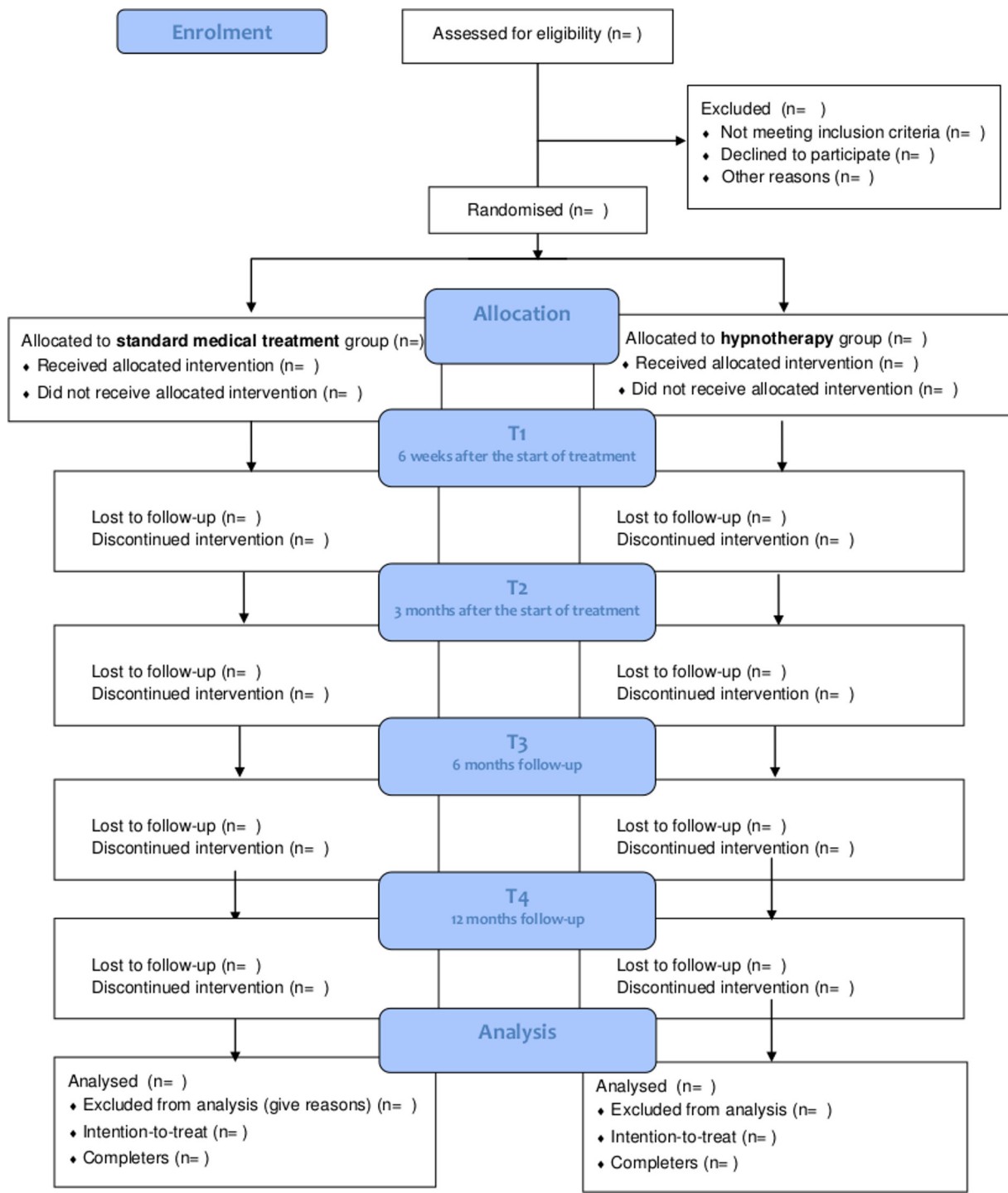

**Figure 2** The Consolidated Standards of Reporting Trials flow diagram indicating the number of participants throughout the study. After eligibility screening, children are randomised in either the hypnotherapy (HT) or standard medical treatment group (SMT). Follow-up measurements take place at 6 weeks after the start of treatment (T1), 3 months after the start of treatment (T2) and 6 months' follow-up (T3) and 12 months' follow-up (T4).

In the first session, an introduction to HT will be given to the child and parents, including an explanation of what HT is and how it may help in reducing symptoms of nausea. Furthermore, the hypnotherapist will take a full history and children and parents are instructed to not talk about the nausea during the treatment period. The hypnotherapist will then start with a breathing exercise and introduce a progressive relaxation, in which children

imagine floating on a big cloud. Positive suggestions to increase the child's belly comfort will also be provided. For instance, the child will be instructed to make hands warm and place both hands on the belly, imagining warmth spreading through their abdomen and especially the stomach.

In the second session, the therapist will repeat the exercise on progressive relaxation. Additionally, the therapist

will introduce an exercise focusing on reduction of anxiety and stress which is called 'the favourite place exercise'.

The third session focuses on ego strengthening and a new exercise will be introduced: 'the rainbow planet exercise' for children attending primary school, and 'the air balloon exercise' for children in secondary school. In the first exercise, children choose a personal need from a rainbow that contains different needs, for example, a healthy stomach, courage, tranquillity or confidence.

In the fourth session, children are encouraged to release stress during the 'the beach without worries exercise' and additional ego strengthening suggestions are made.

The fifth session focuses on reduction of anxiety, stress and ego strengthening, as well as improved functioning of the digestive system. For the digestive system, children visualise a well working digestive system with food sliding through the stomach and bowel in a comfortable way.

In the sixth session, the previous sessions will be evaluated, remaining gastrointestinal problems may be addressed and preceding exercises may be repeated, if requested by children. If no improvement has taken place, an exercise will be introduced in which the child is instructed to look inside the stomach to see 'what the stomach needs'.

After the first session, all children will receive a compact disc containing standard scripts of the exercises used during the sessions. The hypnotherapist will advise children to self-practice these exercises on a daily basis. Additionally, the therapist will encourage children to practice breathing exercises a few time a day.

### SMT+supportive therapy

In the SMT group, children will visit their treating paediatrician six times over a 3-month period. All paediatricians will be instructed to use the same protocol for treating symptoms of nausea. The protocol consists of a stepwise approach. In the first step, children and parents will be educated about CIN and FD, reassured that there is no structural organic underlying disease present, and dietary and lifestyle advices will be provided. Children will be advised to adhere to national practical guidelines for healthy eating by the Netherlands Nutrition Centre. Children will be recommended to avoid products containing caffeine, strong spices, citrus fruits, onions, fatty foods and, if applicable, to stop smoking. Additionally, the paediatrician will explore, together with children and parents, possible connections between stressful moments, emotional problems and complaints of nausea. If connections are present, the paediatrician will encourage children and parents to improve coping strategies to effectively manage stress, to reduce external stressors and to ensure an optimal environment with sufficient relaxation. Children will also be encouraged to continue their normal daily and sport activities and to go to school, to prevent or decrease avoidance behaviour.

In case this does not result in adequate relief of symptoms, the paediatrician continues with the second step.

In the second step, proton pump inhibitors (PPI) in combination with domperidone will be prescribed. If this treatment is not effective in reducing symptoms the paediatrician continues with the third step which includes the prescription of ondansetron and (dis)continuation of PPIs. If again no adequate improvement occurs, in the fourth step, Iberogast will be started for children >12 years. If children do not respond to Iberogast (>12 years) or ondansetron (<12 years), the paediatrician will continue with the fifth step and prescribe erythromycin. Finally, if previous treatment did not prove to be successful in reducing symptoms, cyproheptadine will be prescribed.

The paediatrician will evaluate each step after two to 4 weeks. All dosages, except for Iberogast, will be prescribed according to www.kinderformularium.nl (Dutch medical guideline for paediatricians).

In addition to the medical treatment, children will receive six half hour sessions of supportive therapy given by the treating paediatrician. In these sessions, the symptom progression will be discussed and patient education will be provided. Moreover, exploration of potential contributing triggers (ie, dietary product, emotional problems and stressful events) will be evaluated together with children and parents. Supportive therapy is added to correct for the patient-therapist time in the HT group.

### Co-interventions

After six sessions of HT, children visit their paediatrician to evaluate the effects of HT and, if considered necessary, to receive additional medical treatment.

### Outcomes

#### Primary outcome

The primary outcome of the RCT is the proportion of patients with at least 50% reduction of their symptoms of nausea compare to baseline at 12 months' follow-up. Children and adolescents report symptoms of nausea at home by using the 7-day diary. To promote retention and complete follow-up, children and parents will be reminded on regular basis, via email and phone calls, to fill in the 7-day diary and other questionnaires.

#### *Seven-day diary*

The 7-day diary is used by children and adolescents to score the severity, incidence and frequency of symptoms of nausea, every day during seven consecutive days.[13 21 22]

SON is assessed by the 'nausea face' analogue scale, a validated tool in the paediatric population.[23] Children rate their degree of nausea on a day using six faces: face 0 indicates no nausea, and face 6 indicates nausea as bad as it can be imaged. Scores on the 'nausea face' scale are transported to a daily 0–10 score. Face 0, no nausea, is scored as 0, face 1 is scored as 2, face 2 is scored as 4, face 3 is scored as 6, face 4 is scored as 8 and face 5 is scored as 10.[23] SON score is calculated by summing up the scores of 7 days, giving a maximum score of 70.[13 20–22]

The *incidence* of nausea is assessed using the 'Nausea Incidence' Scale (NIS), adapted from the 5-point dyspepsia

Likert scale.[24 25] It measures the incidence of symptoms during a day, where score 0 for no nausea, score 1 for 1–2 times a day, score 2 for 3–5 times a day, score 3 for intermittent symptoms and score 4 for symptoms were always present. The total sum of the scores of 7 days indicates the severity of the nausea during a week, as experienced by the child. The maximum total score is 28.[13 20–22]

The *frequency* of symptoms of nausea is recorded in minutes/hours per day and is scored by children as 0 when there was no nausea, one if children had <10 min of nausea, 2 for 10–30 min of nausea, 3 for 30 min–2 hours of nausea, 4 for 2–4 hours of nausea and 5 if the nausea lasts >4 hours a day. The 'Nausea Frequency Score' (NFS) is calculated by summing the scores of the 7 days, with a maximum of 35.[13 21 22] Treatment success is defined as at least 50% reduction in the SON, NIS and NFS.

### Secondary outcomes

In addition to the primary outcomes, the present study investigates secondary outcomes, including abdominal pain, dyspeptic symptoms, health-related QoL, anxiety, depression, school absences, parental absence of work and healthcare costs. Secondary outcomes are measured at 6 weeks and 3 months after treatment, and at 6 and 12 months' follow-up after the end of treatment (figure 1).

#### Abdominal pain

A 7-day diary is used to assess the severity and frequency of abdominal pain, every day during seven consecutive days. It is composed of two subscales: the Abdominal Pain Intensity Subscale (APIS) and Abdominal Pain Frequency Subscale (APFS). The APIS will be scored using an affective facial scale ranging from face 0 indicating 'no pain at all' to face 5 indicating 'the most severe pain'. No abdominal pain is scored as 0, faces 1–2 are scored as 1, faces 3–4 are scored as 2 and face 5 is scored as 3. The scores of 7 days are summed up, with a maximum score of 21.[13 20–22] The APFS is recorded in minutes/hours of abdominal pain per day, with score 0 indicating no pain, score 1 if children had <10 min of pain, 2 for 10–30 min of pain, 3 for 30 min–2 hours of pain, 4 for 2–4 hours of pain and 5 for >4 hours of pain. The scores the APFS are summed up, giving a pain frequency score of maximum 35.[13 20–22]

#### Dyspeptic symptoms

Severity and incidence of dyspeptic symptoms is measured using the 5-point dyspepsia Likert scale, previously used in the paediatric population.[24 25] The dyspepsia Likert scale consists of 8 gastrointestinal dyspeptic symptoms: epigastric pain, upper abdominal discomfort, retrosternal pyrosis, sour-bitter taste, halitosis, belching, nausea and early satiety. Children score the *severity* of each symptoms during the previous 2 weeks on a 5-point Likert scale: score 1 'no complaints at all', score 2 'little complaints', score 3 'moderate complaints', score 4 'quite a lot of complaints' and 5 'serious complaints'.[24] A higher sum score indicates more severe dyspeptic complaints (SDC score), with a maximum of 40.

Children report on the *incidence* of each of the symptoms during the previous 2 weeks by scoring: 1 'no complaints', 2 '1–2 times a week', 3 '3–5 times a week', 4 'intermittent complaints' and 5 'complaints were always present'.[24] The Dyspepsia Severity Score is calculated by summing up the scores, giving a maximum value of 40.

#### Health-related QoL

The KIDSCREEN-52 questionnaire measures health-related QoL in children and adolescents. The questionnaire has been shown a valid tool in the Dutch paediatric population.[26 27] The KIDSCREEN-52 consists of items on ten dimensions related to QoL on a 5-point Likert scale: moods and emotions, self-perception, relations with parents and home life, autonomy, physical well-being, psychological well-being, school environment, social support and peers, social acceptance (bullying) and financial resources. For each individual dimension, Rasch scores are computed from the individual items. These are then transformed into T-values: higher T-values indicate a better health-related QoL and well-being.

#### Anxiety and depression

Anxiety and depression are evaluated using the Revised Anxiety and Depression Scale-short version (RCADS-25). This questionnaire has been previously validated in the Dutch paediatric population.[28] The RCADS-25 consists of five subscales measuring symptoms of generalised anxiety disorders, separation anxiety disorder, social phobia, panic disorder and major depressive disorder. Each subscale contains five items and scales range from 0 (never) to 3 (always). The total score on anxiety or depression is the sum of the items measuring symptoms of anxiety and depressive symptoms, respectively. Higher scores indicate more symptoms of anxiety or depression.

#### Cost-effectiveness/cost–utility

The Health Utility Index Mark 3 (HUI) will be used in the cost–utility and cost-effectiveness analysis. The HUI-3 is a multiattribute utility measure of health status in children as reported by parents. Proxy measurements of parents for health status of children are justifiable, as some children may be too young to provide reliable and valid information about their own health status.[29–31] The questionnaire consists of eight dimensions of health status: vision, hearing, speech, ambulation, dexterity, emotion, cognition and pain, with scales varying from highly impaired to normal. Health utilities of 1 indicate perfect health, whereas 0 indicates death. The quality-adjusted life years (QALY) will be calculated by multiplying the sum of the utility of health states by the time in between measurements.

#### Work absenteeism by parents and school absenteeism by children

An adapted version of the Dutch Health and Labor Questionnaire will be used to measure work absenteeism by parents, school absenteeism by children and indirect costs of healthcare utilisation.[32] This adapted version contains three items. Parents indicate whether their

child has been absent from school due to symptoms of nausea, and if yes, the amount of hours per week. For work absenteeism by parents, parents indicate the number of hours they worked less on average because of their child's symptoms of nausea. For the indirect costs of healthcare utilisation, parents indicate additional costs they had due to symptoms of nausea of their child over the past 4 weeks.

### Somatisation

The Children's Somatization Inventory (CSI) measures the extent to which children and adolescents experience somatic symptoms. The questionnaire has been shown a valid and reliable self-report instrument in the paediatric population.[33] The CSI consists of 35 items and on a 5-point Likert scale (0=not at all to 4=a whole lot) and children rate the extent to which they experienced somatic symptoms in the previous 2 weeks. The total score is calculated by summing up the 35 items, with higher scores indicating higher intensity of somatic complaints experienced by the child.

In order to calculate a separate CSI score for non-gastrointestinal (non-GI) symptoms, seven items on GI symptoms (nausea, constipation, diarrhoea, epigastric and abdominal pain, vomiting and bloating) are left out. The total score of somatic symptoms without GI-symptoms is calculated by summing up the scores of non-GI symptoms, with higher scores reflecting higher intensity of non-GI somatic complaints.

### Adequate relief

Parents and children will be asked whether adequate relief of symptoms of nausea has occurred, using a dichotomous scale (yes/no). Adequate relief has been previously used as an endpoint in clinical trials assessing HT in children and adolescents[20] and has been shown a valid outcome measure for functional gastrointestinal disorders.[34]

### Sample size calculation

The primary outcome of the RCT is the proportion of patients with at least 50% reduction of their symptoms of nausea compare to baseline at 12 months follow-up. Based on our pilot study (Vlieger, A. M. 'A pilot-study of hypnotherapy as a treatment for functional nausea in children') and the success percentages in studies using HT in adults with FD[35] and in paediatric patients with cancer,[14] we expect that 80% of the children in the HT group will have >50% reduction of their symptoms of nausea after 1 year. In the SMT group, we anticipate that 50% of the children will have >50% reduction of their symptoms of nausea after 1 year. Based on these expected proportions, 45 children per group will be needed to achieve a power of 80% with a one-sided significance level of 5%. Accounting for a 10% dropout, 100 children will be included in this study. If a child is prematurely withdrawn from the study, he/she will not be replaced; data will be analysed according to the intention to treat (ITT) analysis.

## Statistical analysis
### Primary outcome

Outcomes will be analysed according to the ITT analysis. For the primary outcome, the $\chi^2$ test will be used to compare the proportions of patients with >50% reduction of symptoms of nausea (ie, severity, incidence and frequency of nausea) after 12 months' follow-up between the two groups (HT vs SMT). For all analysis, the significance level for statistical analysis is set at α=0.05. Multiple imputation will be applied to deal with cases of missing data.

### Secondary outcomes

For the secondary outcomes, including the potential influence on abdominal pain, dyspeptic symptoms, health-related QoL, anxiety, depression, work absenteeism by parents, school absenteeism by children, somatisation and adequate relief, the Student's t-test will be used for means of normally distributed data, the Mann-Whitney U test for non-parametric data and the $\chi^2$ test to compare proportions. To calculate the cost-effectiveness, cost–utility and cost-effectiveness ratios will be calculated for the extra costs per child with >50% reduction of symptoms of nausea and the extra costs per QALY.

As secondary analysis, the proportion of patients with >50% reduction of their nausea (severity, incidence and frequency) after treatment, 6 months' and 12 months' follow-up will be compared between groups using multivariate logistic regression correcting for age and centre.

## AMENDMENTS

Prior to implementation, amendments will be examined and approved by the Medical Ethics Committee (MEC). The sponsor will only record non-substantial amendments.

## DATA MONITORING

HT is usually well tolerated in children, without significant side effects. In our previous trials only a minority of children reported some dizziness, mostly during or directly after the end of the first session.[13 21] In case children experience dizziness, they will be advised to execute the remaining sessions in a sitting position instead of a supine position. Children assigned to the SMT group will receive standard medical treatment, including drugs that have been either registered for children, or of which side effects are limited and well known. For these reasons, no Data Monitoring Safety Board will be established.

Study auditing will be accomplished by periodic visits to the participating centres, and by email and telephone contact with local investigators, to ensure the study protocol is being complied with and to discuss any problems that might have arisen.

## POTENTIAL HARMS

In accordance to the legal requirements in the Netherlands (article 10, subsection 1, WMO), the investigator will

inform the subjects and the reviewing accredited METC if harmful events occur. When there are indications that the disadvantage of participation may be significantly greater than was described in the research proposal, the study will be suspended pending further review by the accredited METC. However, the study will not be suspended if it would jeopardise participating children's health.

## ANCILLARY AND POST-TRIAL CARE

In accordance with Article 7 WMO and the Measure regarding Compulsory Insurance for Clinical Research in Humans of 23 June 2003, the sponsor has liability insurance for any damage to children which might emerge from study participation.[36]

## DATA STORAGE

The related information on paper, including 7-day diaries, will be securely stored in a locked file cabinet with limited access. Online questionnaire data will be securely stored using the University's password-protected access systems. Only the main researchers will be given full access to the questionnaire data. All records that contain names will be saved in one file, which will be password protected and only accessible to the main researchers.

## DISSEMINATION POLICY

The researchers will communicate trial results to the public, healthcare providers and other relevant groups via reports, and by publishing in peer-reviewed journals. Negative as well as positive results will be published. The results will be shared with participating children and parents after completion of the trial. All authors who provided substantial contributions to the conduct, interpretation and reporting of the results will be granted authorship on the final trial report.

## DISCUSSION AND CONCLUSION

Chronic nausea is a highly disabling symptom for children with CIN or FD, and poses a risk for negative health outcomes and decreased psychosocial functioning.[2–4] To date large randomised placebo controlled trials evaluating the effect of any drug in children with either CIN or FD are lacking.[37] Current medical treatment is experienced based, however these treatments are symptomatic and often used for months or years.[7 8] For these reasons, new effective treatment options to reduce nausea in children with CIN or FD are warranted.

There are indications that HT can decrease symptoms of functional nausea and dyspepsia in adults,[35] and functional abdominal pain (FAP)[13] and chemotherapy induced nausea in children.[14] Calvert *et al*[35] found that adult patients with FD receiving 12 sessions of HT had significantly less dyspeptic symptoms (59%, n=26) compared with patients receiving medical treatment (33%, n=29) (p=0.02).[35] These

beneficial effects were maintained for more than a year: 56 weeks after the first treatment, 73% of the patients in the HT group reported symptom improvement compared with 43% in the medical treatment group (p<0.01). In children with FAP, Vlieger *et al* found that HT was highly superior compared with SMT to reduce abdominal pain. At 1 year follow-up, 85% of the children in the HT group (n=26) were in clinical remission compared with 25% of the children in the SMT group (n=24) (p<0.001).[13] Additionally, a systematic review including six RCTs evaluating the effectiveness of HT to reduce chemotherapy-induced nausea found HT was most effective when compared with SMT to reduce complaints (D=0.99).[14]

The present study is the first study to investigate the effectiveness of HT on symptoms of nausea in children and adolescents diagnosed with CIN or FD, according to the Rome IV criteria. If shown effective, it may provide an additional treatment option for children with CIN or FD.

The study has several strengths. The first strength is that paediatricians from eleven different hospitals throughout the Netherlands will recruit all children and adolescents. The hospitals, both an academic centre and teaching hospitals, serve an ethnic and socio-economic diverse population of children and adolescents. This recruitment method has two advantages: first, it may reduce response bias to the intervention. It has previously been reported that patients from primary and secondary level care may have different responses to treatment.[38] All children included in the present study receive secondary level care. The second advantage is that the multicentre design of the study will increase generalisability of the trial outcomes.

Another strength of the study is the long-term follow-up of 1 year which allows us to properly compare the potential effectiveness of HT with SMT. It has been known that the severity of functional gastrointestinal symptoms in children varies over time[39] and a continuous improvement in symptoms is often reported in children receiving HT.[13]

This study also has several limitations. The first limitation is that children, parents, investigators and healthcare providers are not blinded for the received treatments, which is not possible due the nature of HT. Several solutions will be applied to limit the risk of bias. First, to minimise performance bias, paediatricians and hypnotherapists will follow treatment guidelines to prevent any use of additional or alternative forms of care during the study period that may influence treatment outcomes. Second, to reduce the risk of detection bias, children and adolescents use reliable outcome measures and record symptoms themselves at home. Moreover, children and adolescents record symptoms of nausea for seven consecutive days, which corrects for individual variability of symptoms over time. Third, the endpoints of the study are preregistered.

The second limitation is that the SMT provided by paediatricians in this study may not reflect usual clinical practice. In the present study, children and adolescents receive longer and more intensive medical therapy from their paediatrician compared with the real medical

practice situation. However, previous studies indicate that patient-provider interactions can largely influence gastrointestinal treatment outcomes.[40] Therefore, it is important to control for the time spent per patient in the SMT group.

If the results of this study show that HT given by a therapist is comparable or slightly more effective than medical treatment provided by paediatricians, HT may become a new treatment strategy to help children with CIN or FD. Furthermore, as HT is presumably less costly than treatments by a specialist it may also decrease healthcare costs.

**Author affiliations**
[1]Department of Pediatric Gastroenterology and Nutrition, Amsterdam UMC, Amsterdam, The Netherlands
[2]Department of Pediatrics, Amphia Ziekenhuis, Breda, The Netherlands
[3]Department of Pediatrics, Maxima Medical Center, Veldhoven, The Netherlands
[4]Department of Pediatrics, Northwest Clinics, Alkmaar, The Netherlands
[5]Department of Pediatrics, Maasstad Ziekenhuis, Rotterdam, The Netherlands
[6]Department of Pediatrics, Zuyderland Medical Center, Heerlen, The Netherlands
[7]Department of Pediatrics, Rijnstate Hospital, Arnhem, The Netherlands
[8]Department of Pediatrics, Haaglanden Medical Center, Den Haag, The Netherlands
[9]Department of Pediatrics, Spaarne Hospital, Hoofddorp, The Netherlands
[10]Department of Pediatrics, Isala Clinics, Zwolle, The Netherlands
[11]Department of Pediatrics, St. Antonius Hospital, Nieuwegein, The Netherlands

**Correction notice** This article has been corrected since it first published online. The open access licence type has been amended.

**Contributors** AMV is the principle investigator, designed the study, wrote the protocol, supervised the trial and supervised writing the manuscript. PDB provided adjustments to the protocol, wrote the manuscript and coordinated the trial. MAB critically revised the protocol, supervised the trial and supervised writing the manuscript. BdH, EMS, HMvW, WTaT, EKG, MG, NB, MMSW, MMvdB, JG, STAT-R, CF participated in patient recruitment and/or treatment, read and approved the manuscript.

**Funding** The Christine Bader Foundation and Gastrointestinal and Liver Foundation supported this work. These funding sources had no role in the design of this study. They will not have any role during the collection of data, analyses or submission of the results.

**Competing interests** None declared.

**Patient consent for publication** Not required.

**Ethics approval** This RCT was approved by the Medical Research Ethics Committees United (MEC-U) in Nieuwegein, the Netherlands (file number: NL51167.100.15). The study will be conducted in accordance with the Medical Research Involving Human Subjects Act (WMO) and conferring to the principles of the Declaration of Helsinki. (64th WMA General Assembly, Fortaleza, Brazil, October 2013). The study will follow the conduct code concerning resistance in minors who participate in clinical trials as defined by the Dutch Pediatric Society. Informed consent will be asked from parents/guardians of children <12 years of age. In children and adolescents 12 years of age informed consent will be asked from the parents-guardians and the children and adolescents. In the event of amendments of the protocol, relevant research ethical committees (RECs) will be informed. Results of the study will be publicly disclosed in a peer-reviewed journal, without any restrictions; both positive as well as negative results will be published.

**Provenance and peer review** Not commissioned; externally peer reviewed.

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
