## [Reviewer comments · BMJ Open]

This paper was submitted to a another journal from BMJ but declined for publication following peer review. The authors addressed the reviewers' comments and submitted the revised paper to BMJ Open. The paper was subsequently accepted for publication at BMJ Open.

(This paper received three reviews from its previous journal but only two reviewers agreed to published their review.)

ARTICLE DETAILS

TITLE (PROVISIONAL)	Gut-Directed Hypnotherapy Versus Standard Medical Care for Nausea in Children with Functional Nausea or Functional Dyspepsia: Protocol of a Multi-Center Randomized Trial.
AUTHORS	Browne, Pamela D; den Hollander, Bibiche; Speksnijder, Esther; van Wering, Herbert; Tjon a ten, Walther; George, Elvira; Groeneweg, Michael; Bevers, Nanja; Wessels, Margreet; van den Berg, Maartje; Goede, Joery; Teklenburg, Sarah; Frankenhuis, Carla; Benninga, Marc; Vlieger, Arine

VERSION 1 – REVIEW

REVIEWER	Rajeev Mohan Kaushik Professor of Medicine Himalayan Institute of Medical Sciences Swami Rama Himalayan University P.O. Jolly Grant-248016 Dehradun Uttarakhand India
REVIEW RETURNED	20-Jul-2018

GENERAL COMMENTS	This randomised controlled trial compares the utility of gut directed hypnotherapy and standard medical treatment in reducing the symptom of nausea in children having chronic idiopathic nausea or functional dyspepsia. The study is first of its kind as these modalities have not been compared directly so far. The study is well designed but certain clarifications are required from the authors. 1. What will be the time interval between two consecutive sessions of hypnotherapy?2. It is not clear that at what point of time, the hypnotherapist will start giving suggestions for achieving the beneficial effect.3. If the hypnotherapist fails in inducing the hypnotic trance by progressive relaxation method, will he/she resort to some other method for inducing the trance?4. As success of hypnotherapy depends upon suggestions given after successful induction of the hypnotic trance, if the child is not able to enter the hypnotic trance, what will be the approach to such a child? Will he/she be excluded from the trial?6. There are certain spelling/grammatical errors like Page 11, line 30. "The incidence nausea" to be modified as "The incidence of nausea". Page 12, line 55. 'complains were always present' to be modified as "complaints were always present". Page 13, lines 46-50. Meaning of the sentence "The questionnaire consists of eight dimensions of health status: vision, hearing,
--

	speech, ambulation, dexterity, emotion, cognition, and pain with 5 or 6 levels per attribute, which scales varying from highly impaired to normal" is not clear. Perhaps the authors mean "The questionnaire....., with scales varying from highly impaired to normal". Page 16, line 47. Kindly check the sentence "...23th June 200th,.....". Page 17, line 30. "Current medical treatment aiming to relieve nausea is experienced based,....." to be modified as "Current.....is experience based,....". 7. Page 17, lines 53-55. The statement "At one year follow-up, 85% of the children in the HT group (N=26) were in clinical remission compared to 85% of the children in the SMT group (N=24) (p<0.001)" is wrong. The quoted reference mentions "At one.....25% of the children in the SMT group".
--	---

REVIEWER	Megan E. Riehl University of Michigan United States of America
REVIEW RETURNED	22-Jul-2018

GENERAL COMMENTS	I believe this protocol and research has the opportunity to provide substantial clinical data pertaining for the non-pharmaceutical treatment of nausea and dyspepsia for children. It is a well conceptualized protocol and has good supporting data from previous pilot study. The authors have clearly addressed common limitations regarding randomized trials and hypnotherapy. I appreciate the use of age appropriate, validated measures pre and post treatment. This is a well constructed study protocol.
---

REVIEWER	Miranda van Tilburg Campbell University, USA
REVIEW RETURNED	27-Oct-2018

GENERAL COMMENTS	Authors propose a much-needed trial for functional nausea with a high likelihood of success. The study design is meticulous and mirrors their previous trials in abdominal pain. BMJ Open publishes study protocols for planned or ongoing studies in order to keep the field up to date and increase research integrity. Completed protocols are not published. This study appears to be 92% completed. The argument that publishing the protocol at this point increases clarity for the field, as well as decrease study deviations of protocol, is unlikely. The initial results of this study are likely planned to be presented at conferences (argument 1) and at this point, any deviations of the original protocol have likely already occurred (argument 2). Can authors indicate what value publishing the protocol this late in their study will add? As for the protocol itself, there are only a few concerns, mostly related to data analyses. (1) Why did the authors decide to give one group only hypnosis and the other group a large number of potential medical and alternative therapies? Why did they not consider standard medical care in both groups (comparing hypnosis to the supportive therapy)?
---

	(2) Chi2 tests are proposed as the main analyses. However, this does not take into account the pre-treatment scores and changes from pre to post-treatment. This may become an issue if pre-treatment scores are not similar in both groups. In addition, the authors should consider controlling for study site and therapist/MD in their analyses, which cannot be done with Chi2 tests. Patients at a site are probably more alike compared to patients at other sites. In addition, therapists/MD may be different in how to apply hypnosis or SMT (again making their patients more alike and other patients different). These type of tests are now proposed in the secondary analyses, not as primary analyses and it is not clear why. (3) It is not clear which is the primary outcome as there are three measures of nausea. Do children need to show 50% reductions on all three or only one of the three? (4) It is not clear why continuous measures (e.g., number of minutes of pain /day) are made categorical. The analysis loses power and results become less intuitive (e.g., it means less to show a change from 1-2 on their categorical measure versus a change from 10 to 30 minutes). Overall a well thought out protocol and I am very interested in seeing the initial results.
--	---

VERSION 1 – AUTHOR RESPONSE

Reviewer 1: Rajeev Mohan Kaushik

This randomized controlled trial compares the utility of gut directed hypnotherapy and standard medical treatment in reducing the symptom of nausea in children having chronic idiopathic nausea or functional dyspepsia. The study is first of its kind as these modalities have not been compared directly so far. The study is well designed but certain clarifications are required from the authors.

We thank the reviewer for the questions. Please find below the answers to these questions.

1. What will be the time interval between two consecutive sessions of hypnotherapy?

Children receive hypnotherapy according to the following schedule: at week 1, 2, 3, 5, 7 and 11. This schedule was similar to our previous RCT on the effectiveness of hypnotherapy by a therapist in comparison with hypnotherapy through self-exercises on CD (Rutten et al., 2014). This schedule has been added to manuscript (section 2.6.1).

2. It is not clear that at what point of time, the hypnotherapist will start giving suggestions for achieving the beneficial effect.

The hypnotherapist starts providing suggestions at the first session (week 1). In this session, the hypnotherapist starts with a breathing exercise to induce a trance like state, and then continues with providing suggestions to increase abdominal comfort and to decrease stress etc.

3. If the hypnotherapist fails in inducing the hypnotic trance by progressive relaxation method, will he/she resort to some other method for inducing the trance?

In children it is relatively easy to induce a hypnotic trance, because children naturally often go in and out of trance during the day. So far, in our hypnosis trials, it was possible to induce a trance in all included children (N=317) (Rutten, 2017; Vlieger, 2007). In teenagers, it may take some extra time before they get into a trance-like state, especially during the first visit. In these cases, the therapist can apply one of the several other techniques available, including eye fixation, eye roll, body scan, countdown breathing, visualization and arm "levitation" techniques.

4. As success of hypnotherapy depends upon suggestions given after successful induction of the hypnotic trance, if the child is not able to enter the hypnotic trance, what will be the approach to such a child? Will he/she be excluded from the trial?

In line with the above, because it is relatively easy to induce a hypnotic trance in children, failure to induce the hypnotic trance has never been a reason to exclude a child in our previous trials (Rutten, 2017; Vlieger, 2007). In these studies, we did observe that if a child/teenager was skeptical about the treatment it occasionally took some extra time to induce a trance like state. Since it is known that, in general, the depth of the hypnotic trance is not correlated with treatment success, in this study it is even accepted when child is in a light trance. In other words, if it is only possible to induce a light trance, the child will remain included in the study.

6. There are certain spelling/grammatical errors like

We thank the reviewer for pointing out these grammatical errors. We incorporated the changes in the new version of the manuscript.

Page 11, line 30. "The incidence nausea" to be modified as "The incidence of nausea". Page 12, line 55. 'complains were always present' to be modified as "complaints were always present".

Page 13, lines 46-50. Meaning of the sentence "The questionnaire consists of eight dimensions of health status: vision, hearing, speech, ambulation, dexterity, emotion, cognition, and pain with 5 or 6 levels per attribute, which scales varying from highly impaired to normal" is not clear. Perhaps the authors mean "The questionnaire....., with scales varying from highly impaired to normal".

Page 16, line 47. Kindly check the sentence "...23th June 200th,.....".

Page 17, line 30. "Current medical treatment aiming to relieve nausea is experienced based,....." to be modified as "Current.....is experience based,....".

7. Page 17, lines 53-55. The statement "At one year follow-up, 85% of the children in the HT group (N=26) were in clinical remission compared to 85% of the children in the SMT group (N=24) (p<0.001)" is wrong. The quoted reference mentions "At one.....25% of the children in the SMT group".

Reviewer 2: Megan E. Riehl

I believe this protocol and research has the opportunity to provide substantial clinical data pertaining for the non-pharmaceutical treatment of nausea and dyspepsia for children. It is a well conceptualized protocol and has good supporting data from previous pilot study. The authors have clearly addressed common limitations regarding randomized trials and hypnotherapy. I appreciate the use of age appropriate, validated measures pre and post treatment. This is a well-constructed study protocol. We want to express our gratitude to the reviewer for reviewing the manuscript.

Reviewer 3: Miranda van Tilburg

BMJ Open publishes study protocols for planned or ongoing studies in order to keep the field up to date and increase research integrity. Completed protocols are not published. This study appears to be 92% completed. The argument that publishing the protocol at this point increases clarity for the field, as well as decrease study deviations of protocol, is unlikely. The initial results of this study are likely planned to be presented at conferences (argument 1) and at this point, any deviations of the original protocol have likely already occurred (argument 2).

1.Can authors indicate what value publishing the protocol this late in their study will add?

We want to express our gratitude for pointing out this remark. We agree with the reviewer that the value of publishing the protocol is diminished concerning argument 1 and 2. However, in our view, publishing this protocol supports another important purpose: to increase transparency in research by preventing publication bias and safeguarding fair use of data. For example, in our recent systematic review, in which we investigated the evidence for pharmacological treatments to treat functional nausea and dyspepsia in children, all included RCTs showed considerable risk of bias, including reporting bias (Browne, 2018). By publishing this protocol prior to publishing the results (the expected date of publication of results is set on March 2020), we aim to contribute to a research environment where publication bias against negative or inconvenient findings is prevented as much as possible. Additionally, by publishing this protocol, we could guarantee clinicians and researchers that no misuse of data analysis took place.

(1) Why did the authors decide to give one group only hypnosis and the other group a large number of potential medical and alternative therapies? Why did they not consider standard medical care in both groups (comparing hypnosis to the supportive therapy)? We agree with the reviewer that providing children hypnotherapy or supportive therapy, in addition to standard medical care, would provide interesting results on the potential additive effect of hypnotherapy. The clinically driven question of this study, however, is whether hypnotherapy, as an independent treatment, could be prescribed to children with chronic nausea in the future instead of long-term use of anti-emetics, which is now often the case. We believe that the current design fits this research question best, and therefore hypnotherapy is studied as an independent intervention, not in combination with standard medical care. Depending on the results of this study, the proposed design by the reviewer could also be interesting to examine in (our) potential future studies.

(2) Chi2 tests are proposed as the main analyses.

However, this does not take into account the pre-treatment scores and changes from pre to post-treatment. This may become an issue if pre-treatment scores are not similar in both groups.

In addition, the authors should consider controlling for study site and therapist/MD in their analyses, which cannot be done with Chi2 tests. Patients at a site are probably more alike compared to patients at other sites.

In addition, therapists/MD may be different in how to apply hypnosis or SMT (again making their patients more alike and other patients different). These type of tests are now proposed in the secondary analyses, not as primary analyses and it is not clear why. In taking the proportion of children with adequate relief of symptoms (defined as 50% reduction of nausea) as a primary endpoint, we followed the recommendation by Saps et al. (2016) on conducting clinical trials in children with Irritable Bowel Syndrome (IBS) (Saps et al., 2016). Although IBS and chronic nausea are different clinical entities, symptoms commonly overlap and therefore we argue that Saps' study (2016) provided relevant recommendations for the design of this study (Friesen, 2016). In Saps' study (2016), the authors recommended to "assess the proportion of patients in each treatment arm who fulfill a pre-established treatment responder definition that represents a clinically meaningful change to the patients". To measure a clinically meaningful change, they recommended calculating the percentage change of symptoms, because absolute change might have different meaning for children with different severity of symptoms. Moreover, to our knowledge, there is no consensus on what includes acceptable levels of change in children with chronic nausea. In this context, we believe that taking the proportion of children with clinically meaningful reduction of symptoms (>50%) will ensure fair specificity in detecting a positive response for adequate relief of nausea. Additionally, calculating the proportion of children with adequate relief increases the clinical relevance of this study for clinicians and patients.

Nonetheless, we agree that the study site and therapist, in some cases, may influence the measured effect. In our previous study on the effect of hypnotherapy on functional abdominal pain in children, no study site or therapist effect on treatment outcome was however observed (Rutten, 2017). The same sites and hypnotherapists take part in this study, and therefore we expect that the measured effect will likewise not be significantly influenced by these factors. In the secondary analyses, logistic regression analysis will be applied to test this hypothesis.

(3) It is not clear which is the primary outcome as there are three measures of nausea. Do children need to show 50% reductions on all three or only one of the three? In this study, and in line with Rutten's study (2014), children need to show 50% reduction on all three measures of the 7-daily diary (i.e. severity of nausea, incidence of nausea and frequency of symptoms) (Rutten et al., 2017). This has been adjusted in the text under section 2.7.1.

(4) It is not clear why continuous measures (e.g., number of minutes of pain /day) are made categorical. The analysis loses power and results become less intuitive (e.g., it means less to show a change from 1-2 on their categorical measure versus a change from 10 to 30 minutes).

We agree with the reviewer that this method may lead to loss of power. However, for children to recall the total duration of their symptoms during the day in minutes is challenging. Thus, to minimize recall

bias, and to follow previous reported studies in children with chronic abdominal pain (in the absence of studies in children with chronic nausea), the minutes of pain were made categorical.

References

- Browne, P. D., Nagelkerke, S. C. J., van Etten-Jamaludin, F. S., Benninga, M. A., & Tabbers, M. M. (2018). Pharmacological treatments for functional nausea and functional dyspepsia in children: a systematic review. *Expert Review of Clinical Pharmacology*, 17512433.2018.1540298. <http://doi.org/10.1080/17512433.2018.1540298>
- Friesen, C. A., Rosen, J. M., & Schurman, J. V. (2016). Prevalence of overlap syndromes and symptoms in pediatric functional dyspepsia. *BMC Gastroenterology*, 16(1), 75. <http://doi.org/10.1186/s12876-016-0495-3>
- Rutten, J. M. T. M., Vlieger, A. M., Frankenhuis, C., George, E. K., Groeneweg, M., Norbruis, O. F., ... Benninga, M. A. (2017). Home-Based Hypnotherapy Self-exercises vs Individual Hypnotherapy With a Therapist for Treatment of Pediatric Irritable Bowel Syndrome, Functional Abdominal Pain, or Functional Abdominal Pain Syndrome. *JAMA Pediatrics*, 171(5), 470. <http://doi.org/10.1001/jamapediatrics.2017.0091>
- Rutten, J. M., Vlieger, A. M., Frankenhuis, C., George, E. K., Groeneweg, M., Norbruis, O. F., ... Benninga, M. A. (2014). Gut-directed hypnotherapy in children with irritable bowel syndrome or functional abdominal pain (syndrome): a randomized controlled trial on self exercises at home using CD versus individual therapy by qualified therapists. *BMC Pediatrics*, 14(1), 140. <http://doi.org/10.1186/1471-2431-14-140>
- Saps, M., van Tilburg, M. A. L., Lavigne, J. V., Miranda, A., Benninga, M. A., Taminiou, J. A., & Di Lorenzo, C. (2016). Recommendations for pharmacological clinical trials in children with irritable bowel syndrome: the Rome foundation pediatric subcommittee on clinical trials. *Neurogastroenterology & Motility*, 28(11), 1619–1631. <http://doi.org/10.1111/nmo.12896>
- Vlieger, A. M., Menko-Frankenhuis, C., Wolfkamp, S. C. S., Tromp, E., & Benninga, M. A. (2007). Hypnotherapy for children with functional abdominal pain or irritable bowel syndrome: a randomized controlled trial. *Gastroenterology*, 133(5), 1430–6. <http://doi.org/10.1053/j.gastro.2007.08.072>

VERSION 2 – REVIEW

REVIEWER	Rajeev Mohan Kaushik Professor of Medicine Himalayan Institute of Medical Sciences Swami Rama Himalayan University P.O. Jolly Grant-248016 Dehradun Uttarakhand India
REVIEW RETURNED	13-Dec-2018

GENERAL COMMENTS	My queries have been answered well. I appreciate the authors for writing this well-designed protocol.
---